# Reproductive Suppression Caused by Spermatogenic Arrest: Transcriptomic Evidence from a Non-Social Animal

**DOI:** 10.3390/ijms24054611

**Published:** 2023-02-27

**Authors:** Baohui Yao, Kang An, Yukun Kang, Yuchen Tan, Degang Zhang, Junhu Su

**Affiliations:** 1College of Grassland Science, Key Laboratory of Grassland Ecosystem (Ministry of Education), Gansu Agricultural University, Lanzhou 730070, China; 2Massey University Research Centre for Grassland Biodiversity, Gansu Agricultural University, Lanzhou 730070, China

**Keywords:** reproductive suppression, delayed testicular development, spermatogenesis, testosterone, AMH, plateau zokor

## Abstract

Reproductive suppression is an adaptive strategy in animal reproduction. The mechanism of reproductive suppression has been studied in social animals, providing an essential basis for understanding the maintenance and development of population stability. However, little is known about it in solitary animals. The plateau zokor is a dominant, subterranean, solitary rodent in the Qinghai–Tibet Plateau. However, the mechanism of reproductive suppression in this animal is unknown. We perform morphological, hormonal, and transcriptomic assays on the testes of male plateau zokors in breeders, in non-breeders, and in the non-breeding season. We found that the testes of non-breeders are smaller in weight and have lower serum testosterone levels than those of breeders, and the mRNA expression levels of the anti-Müllerian hormone (AMH) and its transcription factors are significantly higher in non-breeder testes. Genes related to spermatogenesis are significantly downregulated in both meiotic and post-meiotic stages in non-breeders. Genes related to the meiotic cell cycle, spermatogenesis, flagellated sperm motility, fertilization, and sperm capacitation are significantly downregulated in non-breeders. Our data suggest that high levels of AMH may lead to low levels of testosterone, resulting in delayed testicular development, and physiological reproductive suppression in plateau zokor. This study enriches our understanding of reproductive suppression in solitary mammals and provides a basis for the optimization of managing this species.

## 1. Introduction

Reproduction is the most essential life-history strategy for animals, and the success and efficiency of reproduction affect species survival, population continuation, and evolution [1,2,3]. Environmental change can lead to changes in organisms’ social status, population size, and the number of mates, causing them to adjust their reproductive activities until conditions are more favorable [1,2]. This inhibition of reproductive development, physiology, and/or behavior due to specific environmental or physiological conditions is known as reproductive suppression [1,2,3]. The occurrence of this suppression is sometimes active (the self-limitation hypothesis) [4], but also occurs in passive situations (the dominance control hypothesis) [5], as well as temporary suppression (e.g., interference with mating) [6] and long-term suppression (e.g., most physiological suppression, in the naked mole-rat (*Heterochephalus glaber*)) [7]. Although many advances have been made in understanding the patterns, processes, and mechanisms of reproductive suppression, these results have predominantly come from social animals.

In social mammals, male reproductive suppression is caused primarily by behaviors [8] that include avoidance of inbreeding in the birth group [3,9], direct interference with mating [6], and infanticide [10], or the inability to find a suitable mate [9]. However, physiological suppression also occurs, and the physiological suppression of reproduction involves endocrine dysfunction of the hypothalamic–pituitary–gonadal (HPG) axis [11]. Physiological suppression is principally manifested in delayed puberty or arrested development of secondary sexual characteristics [12,13], impaired or delayed gonadal and gametic development [7,14,15], decreased reproductive hormone levels [10,16], and changes in the molecular environment (i.e., in the expression of reproduction-related genes) [17,18,19]. Some genes are downregulated in subordinate males compared to the dominant male naked mole-rats; for example, genes (*PRM1*, *PRM2*, *ODF3*, and *AKAP4*) involved in sperm maturation at the post-meiotic stage [17]. Genes involved in metabolic and energy-related processes, including lipid biosynthesis, redox processes, and steroid metabolism, as well as genes involved in endocrine signaling (*SSTR3*, *TAC4*, *PRDX1*, and *ACPP)*, are downregulated [18]. Genes (*CYP11A1*, *ABCG8*, and *SCARB1*) involved in steroid hormone biosynthetic pathways are downregulated [19]. These diverse phenomena and mechanisms of reproductive suppression have been reported in social animals, with only isolated reports in solitary animals [2]. Revealing the mechanisms of reproductive suppression in more species would be essential to enriching both the theory and practical management of reproductive regulation.

The plateau zokor (*Eospalax baileyi*) is a typical solitary subterranean rodent in the Qinghai–Tibet plateau (QTP), along with naked mole-rats belonging to Spalacidae [20]. Breeding occurs once a year in groups during the breeding season (mid-April to mid-June), while they live alone during the non-breeding season [21]. The plateau zokor acts as an ecological engineer at a natural population density [22]. As their population density increases, excessive underground excavation causes further damage to vegetation and soil, resulting in the plateau zokor being regarded as a pest [23]. Control practices, such as poison and arrow trap capture [24], have caused the zokor age structure to become younger, so more young individuals had the opportunity to reproduce [25]. These control measures have disrupted the original social structure of the plateau zokor, which may also affect population regulation and reproduction. After some individuals were removed, the remaining individuals’ reproductive opportunities increased. For males, the interference of dominant males with subordinate males was reduced when dominant males were removed. The population density decreased and the reproductive competition among males decreased. The territories, food resources, and nutrition became better, expanding males’ opportunities to find females. Likewise, opportunities for females to exert stricter mate choice also increased.

Due to the high energy cost of cave dwelling underground, the cost of finding a mate and burrowing in subterranean species may be higher compared to surface species, potentially leading to variation in reproductive skew like that of naked mole-rats [26]. Surveys of plateau zokor breeding rates revealed that some adult males in undisturbed sample sites (sites of no control measures for plateau zokor) did not participate in breeding, and paternity analysis of offspring DNA found that 2–4 dominant males sired most of the offspring in a colony (12–18 per colony). Therefore, plateau zokors may exhibit reproductive suppression. We also found that testes size became larger with increasing body weight during the breeding season, but some adult males had smaller, undeveloped testes during the breeding season. Moreover, in the disturbed sample sites (sites with control measures for plateau zokor), the reproductive rate of plateau zokors was higher than in undisturbed sample sites [25]. Thus, reproductive suppression in plateau zokors may be physiological. Previous studies have reported genomic and transcriptomic analyses of reproductive repression in social species e.g., [17,18,19], but studies of solitary mammals are rare [2,4,6]. In addition, reproductive suppression was disturbed by improper control measures, which made species management more difficult.

In this study, we analyzed testicular size, morphology, hormone levels, and RNA-seq transcriptome of male breeders and non-breeders during the breeding season. We collected testes of plateau zokor in the non-breeding season as a control. We aim to reveal the mechanism of physiological reproductive suppression by detecting differences in hormones, testicular development, spermatogenesis, and transcriptome in male plateau zokors. The study of reproductive suppression in solitary mammals will enrich the theory and case studies of reproductive suppression, allowing us to better understand the reproductive strategy, population regulation and population maintenance mechanisms, and management best practices of plateau zokor.

## 2. Results

### 2.1. Morphologic and Histologic Changes in the Testes of Plateau Zokors

We found that the external genitalia and male reproductive tract in male breeders (BSB), non-breeders in the breeding season (BSA), and non-breeding-season males (NBS) were normal, with the size being the only difference. There was a significant decrease in testicular weight (F = 99.62, *p* = 0.00) and testicular coefficient (F = 81.60, *p* = 0.00) in BSA and NBS compared to BSB. At the same time, there were no differences in testicular weight and testicular coefficient between BSA and NBS (*p* > 0.05) (Figure 1A,B). Observation of H&E staining sections then revealed a great many germ cells at all levels in the testes of BSB within the seminiferous tubule, including spermatogonia, primary spermatocytes, secondary spermatocytes, round spermatozoa, and long spermatozoa; whereas in BSA and NBS, only spermatogonia, Sertoli cells, and Leydig cells could be observed in the testes (Figure 1C–E).

### 2.2. Serum Hormone Difference of Plateau Zokors

Serum GnRH (F = 8.78, *p* = 0.004) was higher in BSB than in BSA and NBS, but only the difference between BSB and NBS was significant (*p* < 0.05) (Figure 2A). Levels of LH (F = 7.06, *p* = 0.011) were highest in BSA; these were significantly different from NBS but not BSB (*p* < 0.05) (Figure 2C). The differences in serum FSH (F = 2.39, *p* = 0.142) levels between BSB, BSA, and NBS were not significant (Figure 2D). BSB serum testosterone (F = 26.44, *p* = 0.001) levels were significantly higher compared to BSA and NBS (*p* < 0.05), whereas there was no difference in serum testosterone levels between BSA and NBS (*p* > 0.05) (Figure 2B).

### 2.3. Differentially Expressed Gene Analysis of the Testes in Plateau Zokors

To investigate the gene expression differences that lead to differences between BSA, BSB, and NBS, we analyzed the RNA-seq data to find differentially expressed genes (DEGs). By considering libraries, a total of 12,975 (upregulated: 5014, downregulated: 7961), 5672 (upregulated: 2369, downregulated: 3303), and 15,327 (upregulated: 8365, downregulated: 6962) DEGs were identified in BSA–BSB, BSA–NBS, and BSB–NBS plateau zokor testes, respectively (Figure 3A). From the cluster diagram, BSA and NBS were clustered into one group, significantly different from BSB (Figure 3B). Among the differentially expressed transcripts, Table 1 shows the ten most highly up and downregulated transcripts in BSA, BSB, and NBS.

### 2.4. Differences in Transcription Factors That Regulate AMH (Anti-Müllerian Hormone) Promoter

AMH is a marker of the action of FSH in the prepubertal testes (Figure 4A). Transcription factors bind to the *AMH* promoter, triggering *AMH* expression and increasing AMH production. The mRNA levels of *AMH* and its transcription factors *SOX9* (F = 12.44, *p* = 0.007), *SF1* (F = 22.50, *p* = 0.001), and *GATA4* (F = 11.00, *p* = 0.010) were significantly higher in the BSA and NBS compared to the BSB (Figure 4B).

### 2.5. Differences in Spermatogenic Stages in the Plateau Zokor

To further analyze which stages of spermatogenesis arrest in plateau zokors, we imported clusters of spermatogenesis-related genes that were specifically expressed in mouse spermatogenesis (mitotic, meiotic, post-meiotic, and somatic clusters) from the Germonline database, and mapped them to the DEGs that were identified in the BSA, BSB, and NBS testes. A total of 2793 DEGs in BSA–BSB (Appendix A), 743 in BSA–NBS (Appendix A), and 2805 in BSB–NBS (Appendix A) were involved in different stages of plateau zokor spermatogenesis. In the BSA–BSB comparison, DEGs of somatic and mitotic clusters in BSA were larger than in BSB, while DEGs in meiotic and post-meiotic clusters were smaller than in BSB. In BSA–NBS, the DEGs of the meiotic and post-meiotic clusters in BSA were larger than in NBS. In BSB–NBS, DEGs of somatic and mitotic clusters in the BSB were smaller than in NBS, whereas DEGs of meiotic and post-meiotic clusters were larger than in NBS (Figure 5A). Furthermore, we mapped the expression level of all genes in these four clusters. In the BSA–BSB group, gene expression was downregulated at the meiotic and post-meiotic stages of spermatogenesis in BSA compared to BSB. In BSA–NBS, gene expression was upregulated in spermatozoa at the meiotic and post-meiotic stages in BSA compared to NBS. In BSB–NBS, gene expression was upregulated at the meiotic and post-meiotic stages of spermatogenesis in BSB compared to NBS (Figure 5B). Finally, to understand the characteristics of the spermatogenesis arrest in plateau zokor, we examined both pre- and post-meiotic gene markers of spermatogenesis. In BSA, BSB, and NBS, the mRNA expression levels of *ZBTB16* (F = 0.36, *p* = 0.714) and *STRA8* (F = 0.56, *p* = 0.601) (pre-meiotic markers) were not significantly different, and *SYCE1* (*F* = 47.93, *p* = 0.000) and *SYCP3* (*F* = 17.49, *p* = 0.003) (meiotic markers) in BSB were significantly greater than BSA and NBS. For the genes *TEKT1* (F = 95.86, *p* = 0.000) and *CATSPER1* (F = 324.38, *p* = 0.000) (post-meiotic markers), BSB was greater than BSA, BSA was greater than NBS, and the difference was significant (Figure 5C).

Furthermore, we investigated the morphological differences in plateau zokor testes of different status using immunostaining analysis. From immunostaining analysis, we observed that ZBTB16 (PLZF) and KIT were expressed in spermatogonia in the BSA, BSB, and NBS testes (Figure 6A,B). SYCP3 was expressed in spermatocytes in the BSB testes. SYCP3 has not expressed in the BSA and NBS testes (Figure 6C).

### 2.6. Functional Enrichment Analyses of Differentially Expressed Genes

To study the correlation between genes differentially expressed in testes and germ-line function, we looked for GO term enrichment in the four clusters (Figure 7). In the somatic cluster, the downregulated genes in BSA–BSB and the upregulated genes in BSB–NBS were significantly enriched in GO terms related to cholesterol biosynthetic process, steroid biosynthetic process, and sterol biosynthetic process. In the somatic and mitotic clusters, the upregulated genes in BSA–BSB and the downregulated genes in BSB–NBS were significantly enriched in GO terms related to apoptotic process (including regulation of apoptotic process and positive regulation of apoptotic process), cell adhesion and response to estradiol. The downregulated genes in BSA–NBS were significantly enriched in GO terms related to regulation of apoptotic process and positive regulation of apoptotic process. In the meiotic and post-meiotic clusters, the downregulated genes in BSA–BSB and the upregulated genes in BSB–NBS were significantly enriched in GO terms related to spermatogenesis (including flagellated sperm motility, cilium assembly, spermatid development, and fertilization), sperm structure (including sperm flagellum, sperm midpiece, sperm principal piece, cilium, and acrosomal vesicle), meiotic cell cycle, and male meiosis.

KEGG enrichment of differentially expressed genes in the four clusters of testes are shown in Figure 8. In the somatic and mitotic clusters, the upregulated genes in BSA–BSB and the downregulated genes in BSB–NBS were significantly enriched in Pathways in cancer, Focal adhesion, Relaxin signaling pathway, MAPK signaling pathway, PI3K-Akt signaling pathway, ECM-receptor interaction, Hippo signaling pathway, and AGE-RAGE signaling pathway in diabetic complications. In the meiotic cluster, the downregulated genes in BSA–BSB and the upregulated genes in BSB–NBS were significantly enriched in Oocyte meiosis, Progesterone-mediated oocyte maturation, Amyotrophic lateral sclerosis, Glycerophospholipid metabolism, Glycerolipid metabolism, AMPK signaling pathway, and Ubiquitin mediated proteolysis.

### 2.7. Validation of Gene Expression by qPCR

Nine genes (*AMH*, *SOX9*, *SF1*, *ZBTB16*, *STRA8*, *SYCE1*, *SYCP3*, *TEKT1*, *CATSPER1*) were selected for verification. The results showed that the change trends of these nine genes detected by qPCR were consistent with those from the RNA-seq data (Figure 9), confirming that the RNA-seq data were authentic. The qPCR validation further improves the reliability of the present study.

## 3. Discussion

By studying the testicular size, morphology, hormones, and transcriptome of plateau zokor breeders, non-breeders, and non-breeding-season males, we found that plateau zokors experience reproductive suppression. Non-breeder plateau zokors had small testes, low testosterone levels, and a unique gene expression profile, with the number and relative expression of testicular differentially expressed genes significantly decreased during meiosis. The results of H&E staining and immunohistochemistry sections showed that the germ cells of the non-breeders were spermatogonia. Thus, the spermatogenic arrest of the non-breeders occurred at the spermatogonia stage. The transcriptomic data obtained in this study will help to elucidate testes development in plateau zokors. Results of reproductive suppression in social mammals include reduced levels of reproductive hormones [10,14,16] and impaired or delayed gonadal and gamete development [7,12,13,15]. For example, compared with dominant naked mole-rats, testicular size was small in subordinate males, testosterone levels were lower in subordinate males, and genes involved in spermatogenesis were downregulated in subordinate males [7,17,18,19]. Our findings are consistent with these results. In this study, we investigated the physiological mechanism of reproductive suppression in solitary animals, which is valuable for understanding the reproductive strategies of mammals.

### 3.1. High AMH Levels and Low Testosterone Levels Likely Caused Delayed Testicular Development in Plateau Zokors

AMH and its transcription factors are involved in pubertal development in humans [27,28]. We found that the mRNA levels of *AMH* and its upstream transcription factors *SOX9*, *SF1*, and *GATA4* were significantly higher in BSA, and verified that all the test males were normal individuals (the external genitalia and male reproductive tract in BSB, BSA, and NBS were normal, with the size only difference). Through H&E staining, it can be observed that the germ cells of the testicular tissue of BSA plateau zokors stayed at the stage of spermatogonia, which is similar to the results of H&E staining of testicular tissue of our NSB plateau zokors, indicating that BSA plateau zokor is similar to NSB plateau zokors. Both testes and accessory gonad tissues of BSA degenerate, but they still have the potential to develop into a mature state during the breeding period and participate in reproduction. AMH is secreted by male fetal testicular Sertoli cells, inhibits the development of the Müllerian duct, and makes the mesonephric duct develop into the male reproductive duct under the action of testosterone. Compared with the level before puberty, testosterone in the testes affects the serum level of AMH through androgen receptors after the beginning of puberty and makes AMH decline continuously throughout puberty [29]. In normal puberty, androgens inhibit AMH expression more than FSH-dependent stimulation. In pubertal delay and defective androgen secretion or sensitivity, the lack of androgen secretion or action results in unrestricted activation of the transcription factors *SOX9*, *SF1*, and *GATA4* of *AMH* [27], thus increasing *AMH* mRNA expression and serum AMH. However, intratesticular administration of AMH caused a decline in serum testosterone concentrations by decreasing the rate of testosterone biosynthesis, confirming that AMH can regulate adult Leydig cells androgen production [30]. High testosterone levels promote testicular development, spermatogenesis, and maturation, and dominant males with higher testosterone levels may have higher-quality semen than subordinate males [29]. After 8 weeks of testosterone injection into male plateau zokors during the non-breeding season, An et al. found that the testicular weight increased by 2.9 times compared with the control group [31]. Immunohistochemical results showed that *Stra8*, *γH2AX*, and *SYCP3* positive cells increased. The results showed that testosterone could promote the differentiation of spermatogonia of plateau zokor in the non-breeding season [31]. Therefore, in this study, high levels of AMH may lead to low levels of testosterone, resulting in delayed testicular development, which physiologically inhibited reproduction in the non-breeder plateau zokor. As previous studies have been conducted to verify *AMH* and its upstream transcription factors *SOX9*, *SF1*, and *GATA4* [28], our conclusions are well supported. In the non-breeding season, the expression of *AMH* was higher compared to the breeding season. AMH in the non-breeding season was also higher compared to the breeding season in the male plateau pika (*Ochotona curzoniae*) [32]. However, in naked mole-rats, there was no significant difference in *AMH* mRNA expression between breeders and non-breeders, according to the supplementary data of Bens et al. [18].

### 3.2. The Spermatogenic Arrest of the Non-Breeders Occurred at the Spermatogonia Stage

In this study, significant differences were found between the top 10 genes up- and downregulated in breeders, non-breeders, and non-breeding-season males. In BSA–BSB, the top 10 upregulated genes were mainly involved in the regulation of the blood–testes barrier (*KLF6*) [33] and inhibition of spermatogonial differentiation (*SHISA6*) [34]. The downregulated genes were involved in spermatogenesis and sperm motility (*TNP* [35], *SPATA3* [36], *CCIN* [37], *SPPL2C* [38], *ATP1A4* [39], and *OXCT* [40]). In BSA–NBS, the top 10 upregulated genes were mainly involved in spermatogenesis (*SYT13* [41], *SCNN1B* [42], *SPATA31D1* [43], *TSPAN8* [44], *PTN* [45]). Downregulated genes were involved in spermatogenesis and sperm motility (*CD38* [46], *FGD2* [47]) and apoptosis (*RRAD*) [48]. In BSB–NBS, the top 10 upregulated genes were mainly involved in spermatogenesis (*GSG1* [49], *ODF1* [50], *SPATA20* [51], *SPATA18* [52], *HSPA1L* [53], *DNAH17* [54], and *ACTL7A* [55]). Downregulated genes were involved in apoptosis (*RRAD* [48] and *IFI27* [56]). Compared with BSB, BSA spermatogenesis-related genes were downregulated, indicating the spermatogenic arrest in BSA.

In this study, we found that the gene expression profiles of breeders, non-breeders, and non-breeding season individuals differed significantly between somatic, mitotic, meiotic, and post-meiotic clusters. The differences at the transcriptome level in non-breeder testes compared to breeders were mainly at the meiotic stage. The *KIT* is a marker for differentiating spermatogonial stem cells in several species including mice and goats [57]. In mice, *STRA8* is expressed in differentiated spermatogonia and pre-meiotic spermatocytes, and *STRA8* knockout mouse spermatogonia can initiate meiosis but fail to complete it [58]. *ZBTB16* (also known as *PLZF*) is expressed in undifferentiated spermatogonia and is generally used as a marker for undifferentiated spermatogonia [59]. *SYCE1* (Synaptonemal Complex Central Element 1) is formed between homologous chromosomes during meiotic prophase and exists only during the first meiotic division [60]. *SYCP3* (Conjugation Complex Protein) is required for the assembly of the conjugation complex and is expressed in human spermatocytes during prophase I of meiosis from spermatogonia to the coelomic phase [61]. *TEKT1* is involved in the development of sperm axonemes and flagella in mice [62]. In humans, *CATSPER1* (Cation Channel Sperm-Associated 1), a plasma membrane protein present in the sperm principal piece, is involved in sperm activation and acrosome reactions [63]. Based on our study of H&E staining and immunohistochemistry sections of the testes of breeders and non-breeders, we confirmed that the spermatogenic arrest of the non-breeders occurred at the spermatogonia stage. ZBTB16 and KIT were expressed in spermatogonia in the BSA, BSB, and NBS testes. SYCP3 was expressed in spermatocytes in the BSB testes, but not in BSA and NBS testes. By verifying the mRNA expression levels of meiosis-related markers, the results showed that pre-meiotic markers were not significantly different in non-breeder testes compared to breeder testes. In contrast, meiotic markers and post-meiotic markers were significantly decreased, confirming that the spermatogenic arrest of the non-breeders occurred at the spermatogonia stage. In our study, the physiological suppression of non-breeder plateau zokors was caused by the delayed testicular development. In contrast, differences in spermatogenesis in both breeding and non-breeding males were mainly at the post-meiotic stage in the naked mole-rat [17]. This was due to non-breeding naked mole-rats being primarily subject to behavioral suppression and spermatogenesis occurring between the breeding and non-breeding seasons. Nonetheless, non-breeding males show impaired post-meiotic sperm maturation [17].

Across mammals, the main reason for the prepubertal testicular size is the number of Sertoli cells. After puberty, testicular volume increases dramatically as spermatogenesis begins, germ cells start to differentiate and increase, and germ cells become more numerous [27,30]. The normal proliferation of Sertoli cells increases testes size, whereas premature cessation of Sertoli cell proliferation due to delayed testicular development results in smaller testes [64]. In our study, the small size and weight of the testes in non-breeders may be due to a lack of germ cell proliferation and premature cessation of Sertoli cells. An et al. [31] found that there was no significant difference in the FSH level of plateau zokor between breeding and non-breeding season, which was consistent with the results of our present study. Insufficient testosterone secretion caused the spermatogenic arrest. Low levels of testosterone in BSA and NBS maintain the survival of germ cells, and high levels of testosterone in BSB regulate spermatogenesis and animal reproductive behavior. Although the levels of LH and FSH in the BSA group were high. However, the non-breeder plateau zokors were in the puberty stage, with small testicles and incomplete testicular development. In addition, genes related to meiotic cell cycle, spermatogenesis, flagellated sperm motility, fertilization, sperm capacitation, and sperm structure in non-breeders were downregulated. Therefore, the testosterone level of the BSA group was low and spermatogenesis was incomplete.

### 3.3. Transcriptional Regulation of Reproductive Suppression in Plateau Zokors

GO analysis of genes from somatic, mitotic, meiotic, and post-meiotic clusters revealed that genes related to cholesterol biosynthetic processes and steroid biosynthetic processes were significantly downregulated in the somatic clusters. In our study, we hypothesize that the downregulation of key genes in steroid biosynthesis processes caused an inadequate supply of energy in the testes of non-breeder plateau zokors. In contrast, genes involved in metabolic and energy-related processes such as lipid biosynthesis processes and steroid metabolism processes were also found to be primarily upregulated in the testes of dominant naked mole-rats, indicating increased energy demand in dominant males [18]. Cell adhesion is associated with the blood–testes barrier, which protects developing germ cells from autoimmune responses and exogenous substances [65]. Genes related to cell adhesion were significantly upregulated in somatic and mitotic clusters. This may prevent the separation of germ cells from the basement membrane and subsequent migration to the lumen of the seminiferous tubules in the non-breeders. Estrogen plays a crucial role in normal testicular development and spermatogenesis. It has been shown that 17β-estradiol acts through ESR1 and GPER to activate the EGFR/ERK/c-Jun pathway, and then induces the expression of apoptosis-related genes in germ cells [66]. In this study, genes related to response to estradiol were upregulated in the non-breeders compared to breeders, suggesting that apoptosis occurred in non-breeder spermatogenesis. Apoptosis is a physiological mechanism of programmed cell death that requires eliminating misplaced or damaged cells by genetic decision. It is essential for the normal formation and maintenance of germ cells in the testes. A large increase in germ cell apoptosis is involved in male idiopathic sterility [67]. In our study, genes involved in apoptotic processes were significantly upregulated in somatic and mitotic clusters in non-breeders, suggesting that apoptosis occurs in testicular germ cells of non-breeders, thereby affecting spermatogenesis. Spermatogenesis is a finely regulated process of germ cell proliferation and differentiation [68]. In this study, genes involved in spermatogenesis were downregulated in non-breeders, affecting normal sperm formation and testicular development. Meiosis was downregulated so that the spermatogenic arrest of the non-breeders occurred at the spermatogonia stage which affected normal sperm production. Apoptotic processes increased, and spermatogenesis decreased in non-breeders compared to breeders. Neither non-breeders nor non-breeding season testes were enriched for spermatozoa, suggesting that mature sperm production was not present in non-breeders. Through H&E staining and immunohistochemistry sections, it can be observed that the germ cells of the testicular tissue of non-breeders stayed at the stage of spermatogonia, which was similar to the results of non-breeding season testes. In conclusion, the spermatogenic arrest of the non-breeders occurred at the spermatogonia stage. Increased apoptosis, decreased spermatogenesis during meiosis, and inadequate energy supply may be manifested in the testes of infertile individuals.

KEGG analysis of genes from the four expression clusters revealed that genes related to pathways in cancer, focal adhesion, MAPK signaling pathway, PI3K-Akt signaling pathway, and Hippo signaling pathway were significantly upregulated in the somatic and mitotic clusters in non-breeder plateau zokors. Upregulation of genes such as *PI3K*, *AKT*, and *FOXO3* in the PI3k/Akt pathway leads to upregulation of *CDKN1B*, which encodes a protein that binds to and prevents activation of the cyclin *E-CDK2* or cyclin *D-CDK4* complexes, thereby controlling cell cycle progression in G1 [69]. Upregulation of *CDKN1B* inhibits testes Sertoli cell proliferation. The Hippo pathway regulates multiple cellular functions, e.g., cell proliferation, apoptosis, migration, and differentiation, inhibits excessive cell proliferation, and stimulates apoptosis during development [70]. It has been found that *MST1* overexpression regulates the apoptotic cascade response of caspases, which further triggers apoptosis. In addition, increased *YAP* activity is usually associated with cell cycle entry and apoptosis inhibition. In the present study, both *MST1* and *YAP* expression were upregulated in non-breeders, suggesting that testicular cells in the non-breeders were apoptotic. In the testes of non-breeders, genes involved in AMPK signaling pathway, Oocyte meiosis, and Progesterone-mediated oocyte maturation were downregulated during meiosis. The AMPK signaling pathway maintains the stability of the blood–testes barrier, and in Sertoli cells is a crucial regulator that provides lactate for the energy metabolism of germ cells and maintains spermatogenesis [71]. In contrast, in the present study, the AMPK signaling pathway was significantly downregulated during meiosis in the testes of non-breeders, suggesting that the blood–testes barrier and energy supply are disrupted in non-breeders. In our study, genes related to the oocyte meiosis pathway were also downregulated during meiosis. Although the oocyte meiosis pathway plays a role in female fertility, genes detected in this pathway also regulate sperm meiosis I and II in male spermatogenesis [72]. Downregulated genes of the oocyte meiotic pathway may thus impair sperm meiosis I and II in non-breeder plateau zokors. Genes related to progesterone-mediated oocyte maturation pathway were significantly downregulated during meiosis, and progesterone is a crucial step in stimulating the spermatogonia stage of spermatogenesis via this pathway [73]. Thus, the downregulation of genes in the progesterone pathway observed in our data may prevent spermatogenesis in non-breeders. We show that the main differences in testicular spermatogenesis between non-breeders and breeders in plateau zokors are associated with genes necessary for the meiotic stage. Compared to breeders, the genes of non-breeders at the meiotic and post-meiotic stages of spermatogenesis were significantly downregulated. The non-breeder cannot express key genes at the meiotic stage, resulting in reduced germ cell numbers, smaller testes, damaged and reduced spermatozoa, and reduced testosterone synthesis. In conclusion, our study of plateau zokor testes showed that non-breeders showed meiosis arrest of spermatogenesis. This can result in reduced and impaired sperm counts, which may account for their inability to participate in reproduction. In this study, we aimed to explore the molecular mechanism of gene expression in the spermatogenesis of plateau zokor testes. The next step will be to verify protein expression and function analysis.

### 3.4. Management Implications

A normal population will have self-regulation of reproduction, and it is impossible to reproduce too much. Like most subterranean rodents, the density of plateau zokor has risen rapidly through the reproduction of survivors after control measures [74]. For subterranean rodents, reproductive suppression plays an important role in population self-regulation. Reproductive suppression may be a normal phenomenon in many species. In plateau zokor, it has been found that the proportion of reproductive suppression is higher in undisturbed sites compared with disturbed sites. This is because the population control measures (capturing, trapping, and poisoning) have altered the original social structure, making the age structure of the colony younger [25,75]. In this situation, the dominant males may be removed, so that the original reproductive suppression disappears, and subordinate males have the chance to reproduce. After population control measures, the average diffusion distance of plateau zokors was constant [21]. Therefore, the rapid recovery of plateau zokor populations after population control measures is not the result of diffusion and migration, but the result of reproductive compensation. One possible speculation is that the occurrence of plateau zokor damage on the Tibetan plateau may be due to human disturbance breaking the phenomenon of reproductive suppression. In the follow-up management, such reproductive suppression should be considered, or new prevention and control measures should be innovated from suppression and recovery to play more ecological roles. As plateau zokors are typically subterranean rodents, their activities generally occur underground and are difficult to observe. We have only been able to determine the reproductive suppression of plateau zokors through gonadal development and reproductive hormone levels. The reproduction and reproductive suppression of plateau zokors requires further research.

## 4. Materials and Methods

### 4.1. Animals

We caught plateau zokors alive with tube traps (Baoji Ludixincheng Co. Ltd., Xi’an, China) during the breeding (end of April 2020) and non-breeding (October 2020) seasons in the alpine meadow–steppe area northeast of QTP in Tianzhu Tibetan Autonomous County, Gansu Province, China (37°19′ N, 102°75′ E). The tube trap is a cylinder with a mechanism at one end that can catch subterranean rodents alive so that the animals are not hurt. We found plateau zokor tunnels, dug into them, and put the tube trap into the tunnel. When plateau zokors passed through the tunnel, they were trapped in the tube trap. We checked the tube traps every 20 min. Generally, the age structure of wild rodents was identified on the basis of their body weight or carcass weight, and they were divided into sub-adult, adult, and elderly categories [25,75]. However, no study has been performed on the age determination of plateau zokors in terms of years. According to speculation and observation, plateau zokors generally live for 7–8 years. In addition, plateau zokors may be long-lived like naked mole-rats and other subterranean rodents [74]. During the breeding period, the plateau zokors were captured in the undisturbed sample plot, and 74 adult male plateau zokors were selected according to the age division of the plateau zokor by Su et al. [75].

Male plateau zokors were first examined for breeding status based on their testes; breeders have relatively larger testes, which can be palpated as a bulge in the inguinal pockets in the abdominal region [26]. Next, euthanized males were dissected to determine if they were breeders based on testicular weight. We found that the testes of 4 adult male plateau zokors did not develop, and the reproductive suppression rate was 5.40% (4/74). To verify the reproductive suppression rate, we found an undisturbed sample site for capture, and 39 adult plateau zokors were captured. We found that two of the captured plateau zokors that were altered had no development of sexual organs, and the reproductive suppression rate was 5.13% (2/39).

Four males were non-breeders in the breeding season (BSA) and 20 were breeders in the breeding season (BSB) from 74 adult male plateau zokors. Another 20 adult males were captured in the non-breeding season (NBS). Thus, four BSA, 20 BSB, and 20 NBS were used for analysis. First, plateau zokors were euthanized under anesthesia with isoflurane inhalation. Secondly, plateau zokors were dissected, blood was collected, testes were weighed, and testicular coefficients (testicular coefficient = testicular weight/body weight × 100%) were calculated. Finally, one testis per animal was placed in a 10% formalin solution for fixation and sectioned for H&E staining. Another testis was immersed in liquid nitrogen and used to extract total RNA. This experimental protocol was reviewed and approved by the Animal Ethics Committee of Gansu Agricultural University (approval number GAU-LC-2020-014), and conducted in compliance with the ARRIVE guidelines.

### 4.2. Hormone Determination

We used enzyme-linked immunosorbent assay kits for GnRH, LH, FSH, and testosterone from Cloud-Clone Corp. (Wuhan, China). Hormone determination and conditions followed those of Kang et al. [74].

### 4.3. H&E Staining and Immunohistochemistry

Testicles were obtained from the 10% formalin solution stored samples, then embedded in paraffin wax. After the embedded was complete, 5 µm sections were sliced using a rotary slicer (Leica RM2255). The sections were respectively stained with hematoxylin and eosin solution. Finally, the sections were dehydrated with a graded ethanol and xylene series, respectively. The sections were sealed with neutral gum before microscopic examination and photography.

Testes sections were deparaffinized and rehydrated with xylene and a graded ethanol series, respectively. Testes sections were soaked in sodium citrate buffer solution for antigen repair, and 3% H_2_O_2_ was used to eliminate endogenous peroxidase activity and washed with PBS. The experimental group was treated with rabbit anti-PLZF (Boaosen, Beijing, China, BS-5971R, 1:250), rabbit anti-KIT (Boaosen, Beijing, China, BS-0672R, 1:250), and rabbit anti-SYCP3 (Boaosen, Beijing, China, BS-106606R, 1:250), and the negative control was treated with PBS. Sheep anti-rabbit IgG and horseradish enzyme-labeled chain albumen working solution was then added, incubated at 37℃, and stained with DAB (Boaosen, Beijing, China, C-0010) and hematoxylin. Then, the sections were dehydrated using an ethyl alcohol series, cleared in xylene, and photographed using Motic Images Plus 3.0 software.

### 4.4. RNA-seq

Three testicular samples each from the BSA, BSB, and NBS groups were selected for transcriptome sequencing and sent to Beijing NovoGene Co., Ltd. (Beijing, China) for sequencing.

### 4.5. Total RNA Extraction, Library Preparation, and Sequencing

Total RNA was isolated from the nine testicular tissues with an RNA Nano 6000 Assay Kit (Agilent Technologies, Santa Clara, CA, USA) following the manufacturer’s protocols. The Agilent 2100 bioanalyzer and NanoPhotometer spectrophotometer were adopted to assess RNA integrity and concentration.

Library preparation. Total RNA was used as input material for the RNA sample preparations. Briefly, mRNA was purified from total RNA using poly T oligo-attached magnetic beads, and cDNA was synthesized using mRNA as a template. Selected cDNA library fragments that were preferentially 370–420 bp in length were purified with the AMPure XP system (Beckman Coulter, Beverly, MA, USA). After PCR amplification, the PCR product was purified with AMPure XP beads, and the library was finally obtained. After construction, the library was initially quantified using a Qubit2.0 fluorometer. qRT-PCR was applied to accurately quantify the effective concentration of the library (higher than 2 nM) to ensure the quality of the library.

Transcriptomic sequencing. After the library was quantified, the different libraries were pooled according to the effective concentrations and the target amounts of data produced and sequenced with the Illumina NovaSeq 6000 machine with 150 bp ends read.

### 4.6. Transcriptomic Data Analysis

Quality control. The image data measured with the high-throughput sequencer were converted into sequence data (reads) using CASAVA base recognition. Raw data (raw reads) in fastq format were first processed through in-house perl scripts, and clean data (clean reads) were obtained by removing reads that contained adapters, reads containing *N* bases, and low-quality reads from the raw data. Q20, Q30, and GC content of the clean data were then calculated. All the downstream analyses were based on the clean, high-quality data.

Read mapping to the reference genome. Reference genome (BioProjects: PRJNA254049; *Nannospalax galili*) and gene model annotation files were downloaded from the genome website directly. An index of the reference genome was constructed using Hisat2 (v2.0.5) and paired-end clean reads were aligned to the reference genome using Hisat2 (v2.0.5).

Novel transcript prediction. The mapped reads of each sample were assembled with StringTie (v1.3.3b) in a reference-based approach.

Quantification of gene expression level. FeatureCounts v1.5.0-p3 was implemented to count the read numbers mapped to each gene, and the FPKM for each gene was calculated based on the length of the gene and read count mapped to the gene.

Differential expression analysis. Differential expression analysis of two conditions/groups (two biological replicates per condition) was performed using the DESeq2 R package (1.20.0). Padj ≤ 0.05 and |log2 (fold-change)| ≥ 1 were set as the threshold for significant differential expression.

### 4.7. GO and KEGG Enrichment Analysis

Gene clusters expressed at somatic (SO), mitotic (MI), meiotic (ME), and post-meiotic (PM) stages of mouse spermatogenesis were obtained from Germonline (v.4.0) [76] and mapped to the DEGs that were identified in the plateau zokor testes, following the methods of Mulugeta et al. [17]. The different stages of spermatogenesis DEGs were classified as upregulated or downregulated and plotted [17,77]. DAVID 6.8 (https://david.ncifcrf.gov, accessed on 14 May 2022) was used to perform GO and KEGG analysis in four gene clusters differentially expressed in testes [78]. Each cluster was matched with enriched GO terms and KEGG pathways that were ordered according to peak expression in SO, MI, ME, and PM clusters [77].

### 4.8. Quantitative Real-Time Reverse-Transcription PCR

In order to validate the expression profile of genes from RNA-seq, we chose nine genes for further quantitative real-time PCR (qPCR) detection. Reverse transcription was performed using Evo M-MLV RT Premix for the qPCR kit (Accurate, Changsha, China). The cDNA was amplified by qPCR using TB Green^®^ Premix Ex Taq™ II (Takara, Beijing, China) in a real-time PCR system (Light Cycler 96 System, Roche). The 2^−ΔΔCq^ method was used to deal with the results of qPCR, and the relative expression level of each gene was corrected using the reference gene. A commercial sequencing system (TsingKe, Xi’an, China) was used to synthesize the primer sequences (Table 2).

### 4.9. Data Analysis

R (v.3.5.2) software was used for statistical analysis [79]. Testicular weight and serum hormones were compared using the aov function. For the comparison of relative gene expression of three sample groups, an ANOVA was applied followed by Duncan’s post hoc test. The threshold for significance was *p* < 0.05.

## 5. Conclusions

We found that the testes of non-breeders are smaller in weight than those of breeders, and high levels of AMH may lead to low levels of testosterone, resulting in delayed testicular development, and physiological reproductive suppression in plateau zokor. Genes related to spermatogenesis are significantly downregulated in both meiotic and post-meiotic stages in non-breeders. Our study uncovered insights into testicular development and spermatogenesis under reproductive suppression in plateau zokors, which are of great value in enriching our understanding of reproductive suppression in solitary mammals.

## Figures and Tables

**Figure 1 ijms-24-04611-f001:**
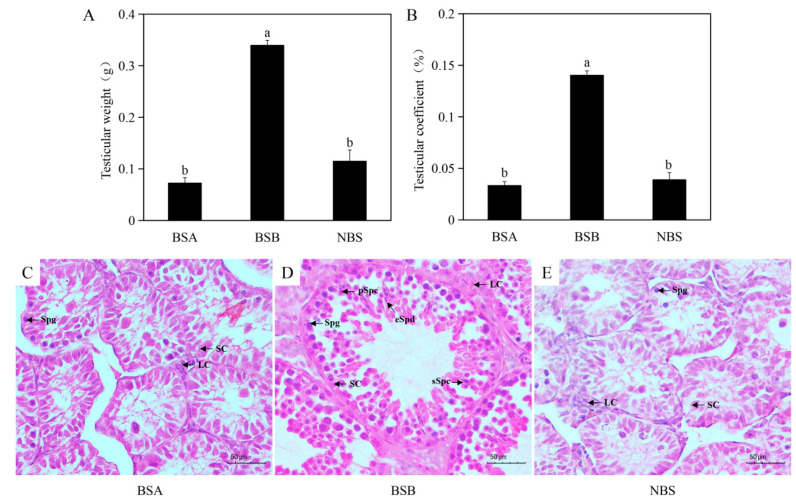
Morphological and histological changes in BSA (*n* = 4), BSB (*n* = 20), and NBS (*n* = 20). (**A**) Testicular weight of the plateau zokor. (**B**) Testicular coefficient of the plateau zokor. (**C**) H&E staining of a representative non-breeder male during the breeding season. (**D**) H&E staining of a representative breeder male during the breeding season. (**E**) H&E staining of a representative non-breeding-season male. SC, Sertoli cell; LC, Leydig cell; Spg, spermatogonia; pSpc, primary spermatocyte; sSpc, secondary spermatocyte; eSpd, elongated spermatid. BSA, non-breeder in the breeding season; BSB, breeder in the breeding season; NBS, non-breeding-season zokors. Groups labeled with the same lowercase letter are not significantly different, *p* > 0.05. Error bars are standard errors.

**Figure 2 ijms-24-04611-f002:**
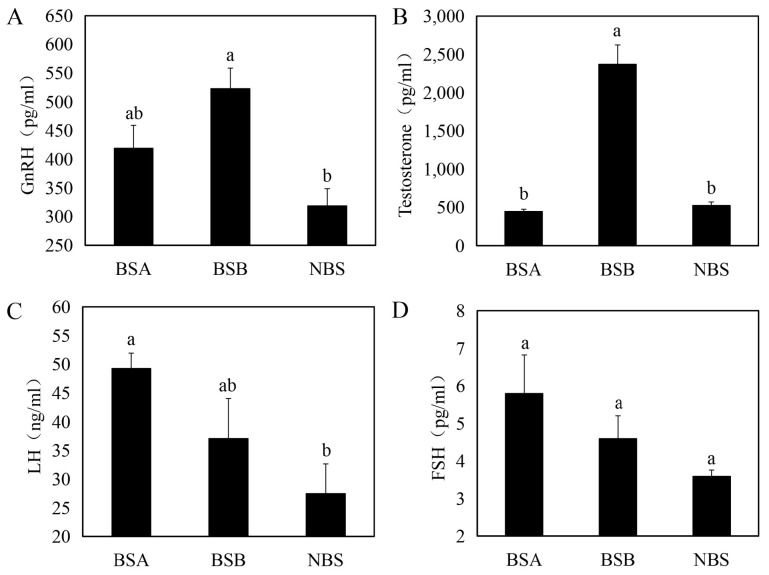
Differences in serum reproductive hormones in BSA (*n* = 4), BSB (*n* = 20), and NBS (*n* = 20). (**A**) GnRH. (**B**) Testosterone. (**C**) LH. (**D**) FSH. BSA, non-breeder in the breeding season; BSB, breeder in the breeding season; NBS, non-breeding-season zokors. Groups labeled with the same lowercase letter are not significantly different, *p* > 0.05. Error bars are standard errors.

**Figure 3 ijms-24-04611-f003:**
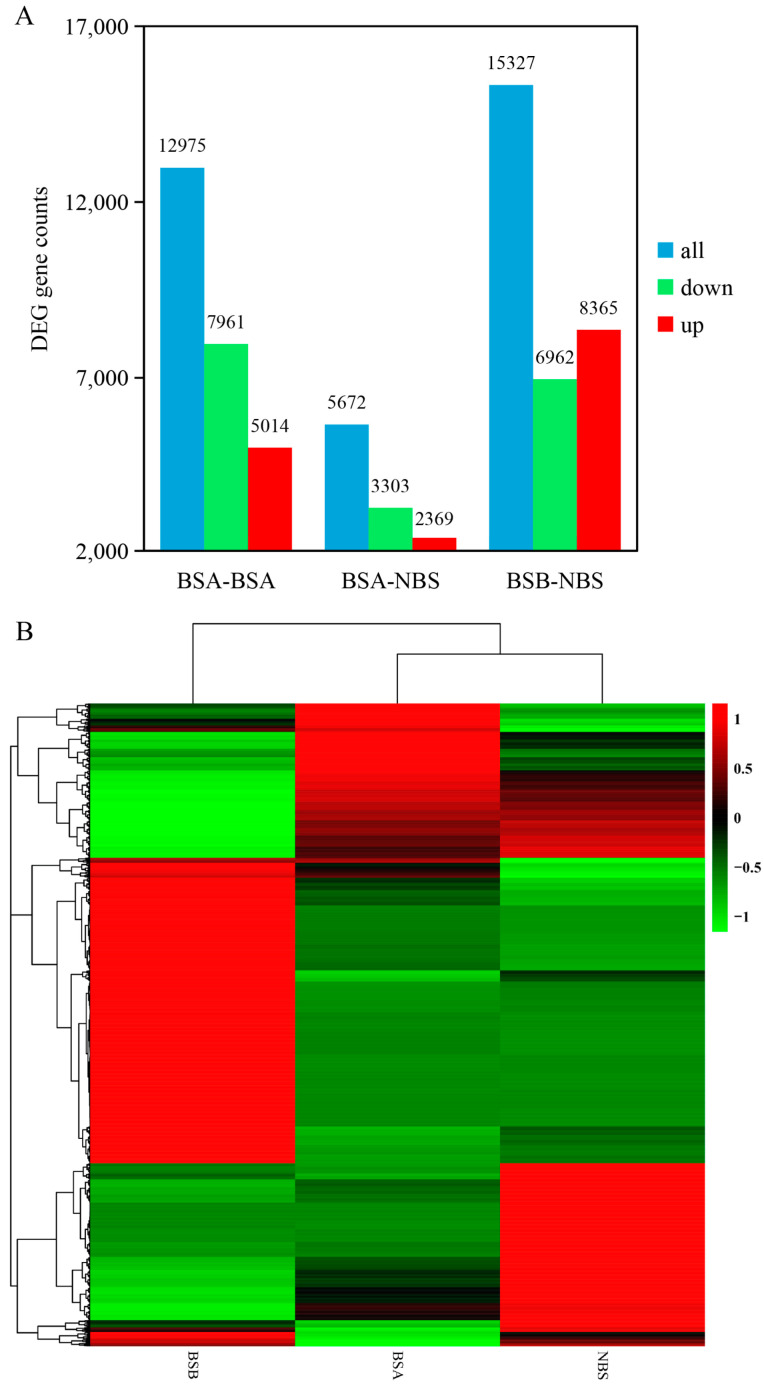
Number of DEGs and cluster maps of BSA, BSB, and NBS. (**A**) Plot of unigenes and number of DEGs in the BSA–BSB, BSA–NBS, and BSB–NBS comparisons. (**B**) Cluster map of BSA, BSB, and NBS. BSA, non-breeder in the breeding season; BSB, breeder in the breeding season; NBS, non-breeding-season zokors.

**Figure 4 ijms-24-04611-f004:**
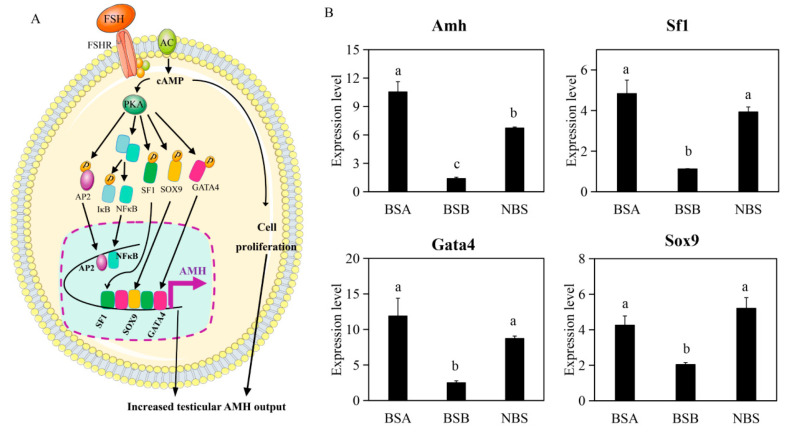
Regulation of AMH production and mRNA levels of transcription factors *SOX9*, *SF1*, and *GATA4*. (**A**) Transcriptional activation of AMH is regulated by FSH in the prepubertal testes [27]. (**B**) mRNA levels of *AMH*, *SOX9*, *SF1*, and *GATA4* in BSA (*n* = 3), BSB (*n* = 3), and NBS (*n* = 3). BSA, non-breeder in the breeding season; BSB, breeder in the breeding season; NBS, non-breeding-season zokors. Groups labeled with the same lowercase letter are not significantly different, *p* > 0.05. Error bars are standard errors.

**Figure 5 ijms-24-04611-f005:**
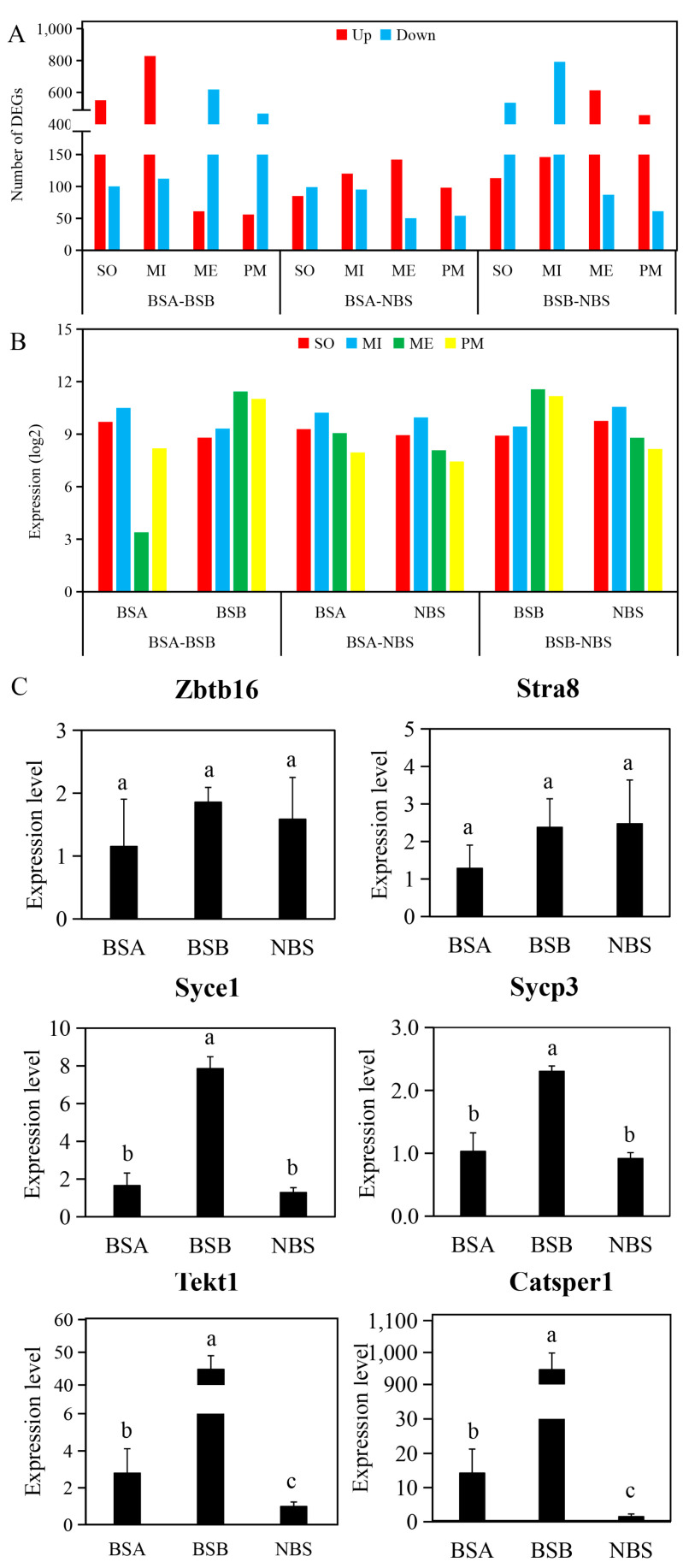
Gene expression profiles of the male testes. (**A**) Bar graph showing the number of plateau zokor DEGs. (**B**) Bar graph showing the average expression levels of all genes. (**C**) The mRNA expression levels of the pre- and post-meiotic markers in BSA (*n* = 3), BSB (*n* = 3), and NBS (*n* = 3). *ZBTB16* and *STRA8* were pre-meiotic markers, *SYCE1* and *SYCP3* were meiotic markers, and *TEKT1* and *CATSPER1* were post-meiotic markers. BSA, non-breeder in the breeding season; BSB, breeder in the breeding season; NBS, non-breeding-season zokors. SO, somatic clusters; MI, mitotic clusters; ME, meiotic clusters; PM, post-meiotic clusters. Groups labeled with the same lowercase letter are not significantly different, *p* > 0.05. Error bars are standard errors.

**Figure 6 ijms-24-04611-f006:**
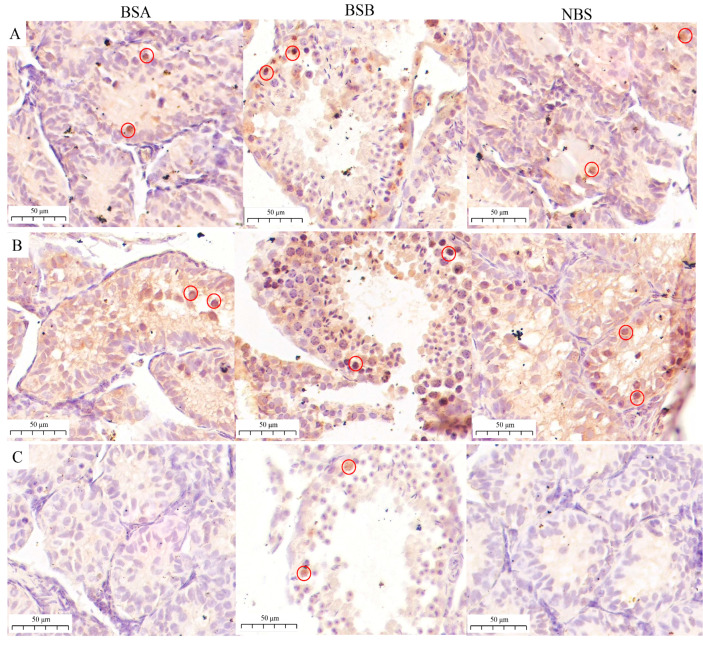
Immunostaining of testes of plateau zokor. (**A**) KIT-positive cells in testes. (**B**) ZBTB16-positive cells in testes. (**C**) SYCP3-positive cells in testes. BSA, non-breeder in the breeding season; BSB, breeder in the breeding season; NBS, non-breeding-season zokors.

**Figure 7 ijms-24-04611-f007:**
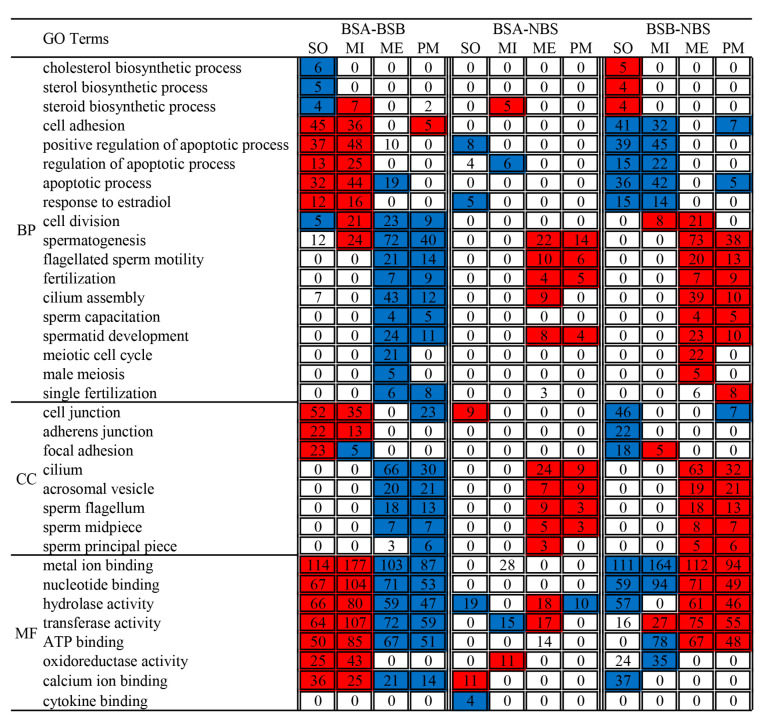
GO term enrichment of differentially expressed genes in four clusters of testes. The numbers of genes associated with a specific GO term and enriched in each cluster are given within rectangles in bold as observed. The color code indicates overrepresentation (red) and underrepresentation (blue). BSA, non-breeder in the breeding season; BSB, breeder in the breeding season; NBS, non-breeding-season zokors. SO, somatic clusters; MI, mitotic clusters; ME, meiotic clusters; PM, post-meiotic clusters.

**Figure 8 ijms-24-04611-f008:**
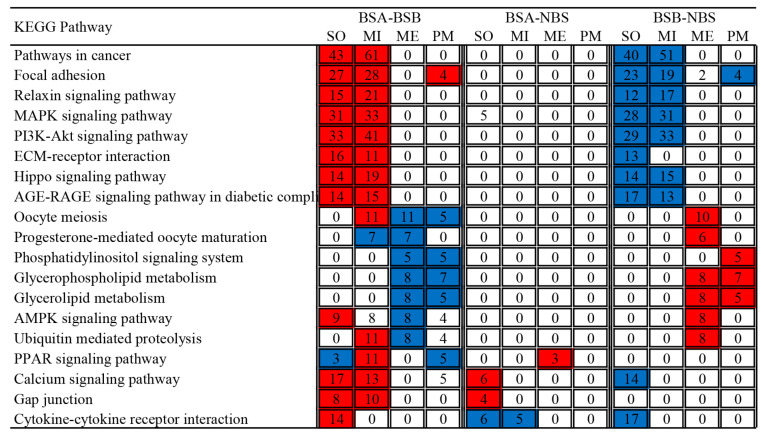
KEGG enrichment of differentially expressed genes in four clusters of testes. The numbers of genes associated with a specific KEGG and enriched in each cluster are given within rectangles in bold as observed. The color code indicates overrepresentation (red) and underrepresentation (blue). BSA, non-breeder in the breeding season; BSB, breeder in the breeding season; NBS, non-breeding-season zokors. SO, somatic clusters; MI, mitotic clusters; ME, meiotic clusters; PM, post-meiotic clusters.

**Figure 9 ijms-24-04611-f009:**
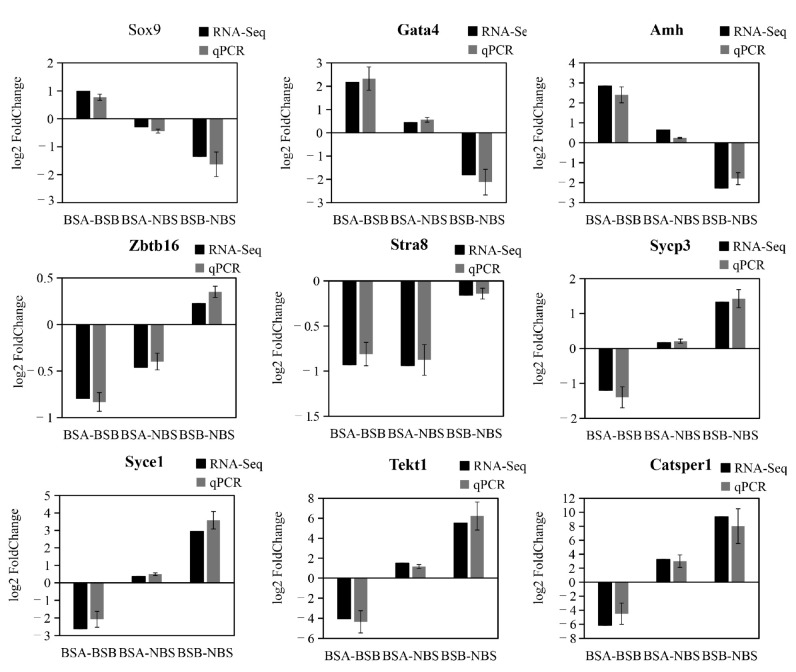
The qPCR validation of genes in RNA-seq data.

**Table 1 ijms-24-04611-t001:** Top ten upregulated and downregulated transcripts in BSA–BSB, BSA–NBS, and BSB–NBS.

	Upregulated			Downregulated		
	Gene Symbol	Gene Name	Fold Change	Gene Symbol	Gene Name	Fold Change
BSA–BSB	Cx3cl1	C-X3-C motif chemokine ligand 1	6.52	Tnp1	Transition protein 1	−9.46
	Col3a1	Collagen type III alpha 1 chain	4.73	4933402P03Rik		−9.27
	Star	Steroidogenic acute regulatory protein	3.90	Oxct2a	Oxct2a 3-oxoacid coa transferase 2a	−8.90
	Shisa2	Shisa family member 2	3.73	Mst1r	Macrophage stimulating 1 receptor	−8.25
	Rprm	Reprimo, tp53 dependent g2 arrest mediator homolog	3.26	Spata3	Spermatogenesis associated 3	−8.04
	Matn2	Matrilin 2	3.25	1700001O22Rik		−7.45
	Prss35	Serine protease 35	2.98	Sppl2c	Signal peptide peptidase like 2c	−7.25
	Klf6	Kruppel like factor 6	2.91	Atp1a4	Atpase Na+/K+ transporting subunit alpha 4	−6.00
	Amh	Anti-mullerian hormone	2.84	Ccin	Calicin	−5.94
	Tubb5	Tubulin beta-5 chain	2.43	Ace	Angiotensin I converting enzyme	−5.61
BSA–NBS	Syt13	Synaptotagmin 13	5.64	Mt-nd6	Mitochondrially encoded nadh:ubiquinone oxidoreductase core subunit 6	−13.23
	Padi1	Peptidyl arginine deiminase 1	5.59	Cd38	Cd38 molecule	−8.05
	Fer1l4	Fer-1 like family member 4	5.45	Mt-nd3	Mitochondrially encoded nadh:ubiquinone oxidoreductase core subunit 3	−7.57
	Scnn1b	Sodium channel epithelial 1 subunit beta	5.22	Sptbn5	Spectrin beta, non-erythrocytic 5	−6.36
	Spata31d1a	Spermatogenesis-associated protein 31d1	4.91	Gbp7	Guanylate binding protein 7	−4.91
	Tspan8	Tetraspanin 8	4.89	Wfikkn2	Wap, follistatin/kazal, immunoglobulin, kunitz and netrin domain containing 2	−4.73
	Dmgdh	Dimethylglycine dehydrogenase	4.87	Ephx3	Epoxide hydrolase 3	−4.41
	Camk1g	Calcium/calmodulin dependent protein kinase Ig	4.50	Pcbd1	Pterin-4 alpha-carbinolamine dehydratase 1	−4.14
	Myoc	Myocilin	4.10	Fgd2	Fyve, rhogef and ph domain containing 2	−4.03
	Ptn	Pleiotrophin	3.68	Rrad	Rrad, ras related glycolysis inhibitor and calcium channel regulator	−3.94
BSB–NBS	Gsg1	Germ cell associated 1	12.95	Tgfbi	Transforming growth factor beta induced	−5.70
	Odf1	Outer dense fiber of sperm tails 1	11.90	Rrad	Rrad, ras related glycolysis inhibitor and calcium channel regulator	−5.66
	Spata20	Spermatogenesis associated 20	10.01	Ifi27	Interferon alpha inducible protein 27	−4.36
	Hspa1l	Heat shock protein family a (hsp70) member 1 like	9.51	Prr5	Proline rich 5	−3.73
	Actl7a	Actin-like protein 7a	9.38	Slc48a1	Solute carrier family 48 member 1	−3.65
	Slc13a5	Solute carrier family 13 member 5	8.76	Tagln2	Transgelin 2	−3.63
	Gm5617	Chromosome 11 open reading frame 71	8.03	Wipf3	Was/wasl interacting protein family member 3	−3.56
	Spata18	Spermatogenesis associated 18	7.61	Stk10	Serine/threonine kinase 10	−3.54
	Kcnt1	Potassium sodium-activated channel subfamily t member 1	6.94	Ankrd46	Ankyrin repeat domain 46	−3.39
	Dnah17	Dynein axonemal heavy chain 17	6.44	3632451O06Rik		−3.20

**Table 2 ijms-24-04611-t002:** List of the qPCR primer sequence.

Gene Symbol	Primer Sequence (5′→3′)	Product Size (bp)
*SF1*	F: GATTCCCCGCAACAACCTCC	193
	R: TTCCTCGTTCACCATCCCAA	
*SOX9*	F: AGTGTAGAGGAGCATTGGTAAGC	168
	R: GCCTTTGCTTGCACTTCGAG	
*AMH*	F: GTGCGCTGCTTCTGCTAAAA	124
	R: GCGCAAGGTGCTTCCGTTA	
*ZBTB16*	F: CCCCGGTGGAGAAGCATTTG	170
	R: CTGAATGAAACCCTGAGGGAGG	
*STRA8*	F: GCACACCGTTTTGAATCCCC	197
	R: TTTCCCGCCATCACGACTTT	
*SYCE1*	F: CAGGTCATCGGCAACTGGGA	151
	R: GGACTTCAATCCGGGGCTCTA	
*SYCP3*	F: TACTGAAGAAAATACTCCAGGTGA	134
	R: TGGCAAGAAGAGCCTTGTTAAT	
*TEKT1*	F: GTGCACGACTGTAACCTCCA	183
	R: CCACACCCCTGCAATGAGAT	
*CATSPER1*	F: CAGGCCACACCATCTTGACT	107
	R: CACCATGTAAGTGAGGCCGT	
*β–Actin*	F: TTGTGCGTGACATCAAAGAG	208
	R: ATGCCAGAAGATTCCATACC	

## Data Availability

RNA-Seq data were deposited in the China National Center for Bioinformation (PRJCA010603, https://ngdc.cncb.ac.cn/gsa/browse/CRA008119, accessed on 8 September 2022). All other study data were included in this article and/or supporting information.

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
