# Peer review of "Reproductive Suppression Caused by Spermatogenic Arrest: Transcriptomic Evidence from a Non-Social Animal"

_ijms, 2023, doi:10.3390/ijms24054611_

Round 1

Reviewer 1 Report

In this study, the researchers tried to understand the mechanism of male reproductive suppression in Plateau zokor, dominant subterranean and solitary rodent in the Qinghai-Tibet Plateau. By performing morphological, hormonal and transcriptomic assays on testis of male plateau zokors in breeders (BSB), non breeders (BSA) and non breeding seasons (NBS), they found that testes in BSA and NBS are significantly smaller than those from BSB due to spermatogenesis defect at pre-meiotic stage. This spermatogenic defects are likely to be caused by the low testosterone levels and AMH upregulation in BSB and NBS. This paper enriches our understanding of reproductive suppression in solitary mammals and is of certain significance. However, several points need to be addressed:

Major concerns:

1.      As the authors stated, one major significance of this paper is that they conducted their study on solitary mammal rather than the well-studied social mammals. Thus, they should carefully compare their results with the findings from social mammals, and then discuss the differences between them.

2.      The authors compared the gene expression in testes from BSB, BSA and NBS only by data from sequencing. This data should be verified by RT-PCR and WB to make sure of accuracy.

3.      The authors analyzed the stage of spermatogenesis defects in BSA and NBS only by analyzing the sequencing data, and concluded that “spermatogenesis ceased completely in the non-breeding season testes and that the testes of the non-breeders underwent partial spermatogenesis”. However, the sequencing data is not sufficient to support the “completely” cease of spermatogenesis. They should stain some markers for spermatogonia, spermatocytes and spermatids on testis sections, and then analyze the result from the IHC staining carefully to see how spermatogenesis occurred in BSA and NBS.

Minor concerns:

1.      In section 3.3, the authors concluded “upregulation of apoptosis and the downregulation of spermatogenesis and meiosis suggest that spermatogenic arrest in the testes of non-breeders may occur as early as the spermatogonial stem cell differentiation stage”, However, they did not find significant difference in the mRNA levels of ID4, STRA8 and ZBTB16 between BSA, BSB and NBS. Thus, this conclusion is not well supported by their own data.

2.      In fig 5, they used 3 and 2 genes as markers for pre-meiotic and post meiotic stages, but only Sycp3 was used as marker for meiotic stage. More markers should be shown for meiotic stage.

3.      Some descriptions in this paper are not appropriate. Such as: “the downregulation of spermatogenesis and meiosis”, “Meiosis was downregulated”. Meiosis and spermatogenesis can not be up or down regulated, only the expression of genes in these processes can be up or down regulated.

4.      Besides, there are a lot of typographical errors.

 Taken together, more experiments and discussions are needed to support the conclusions and improve the significance of this paper. Thus, this paper should be revised thoroughly.

Author Response

Major concerns:

1- As the authors stated, one major significance of this paper is that they conducted their study on solitary mammal rather than the well-studied social mammals. Thus, they should carefully compare their results with the findings from social mammals, and then discuss the differences between them.

Response 1: Thank you for your insightful comment. We have compared our results with the findings from social mammals and made the necessary changes in the Discussion section. (Line 247-252, 332-337).

Results of reproductive suppression in social mammals include reduced levels of reproductive hormones (Bennett, 1994; Cant, 2000; Van den Berghe et al., 2012) and impaired or delayed gonadal and gamete development (Clutton-Brock, 1998; Maggioncalda et al., 2002; Charpentier et al., 2008; Kruczek and Styrna, 2009). For example, compared with dominant naked mole-rats (Heterochephalus glaber), testicular size was small in subordinate males, testosterone levels were lower in subordinate males, and genes involved in spermatogenesis were downregulated in subordinate males (Clutton-Brock, 1998; Mulugeta et al., 2017; Bens et al., 2018; Sahm et al., 2021). Our findings are consistent with these results. In contrast, differences in spermatogenesis of both breeding and non-breeding males were mainly at the post-meiotic stage in the naked mole-rat (Mulugeta et al., 2017). This is due to non-breeding naked mole-rats being primarily subject to behavioral suppression and spermatogenesis occurring between the breeding and non-breeding seasons. Nonetheless, non-breeding males showed impaired post-meiotic sperm maturation (Mulugeta et al., 2017). In our study, the physiological suppression of non-breeder plateau zokors was caused by the delayed testicular development. Spermatogenic arrest in the testes of non-breeder may occur at the spermatogonia cell stage. (Line 248-253, 333-339).

Clutton-Brock, T.H. Reproductive skew, concessions and limited control. Trends Ecol. Evol. 1998, 13, 288–292. https://doi.org/10.1016/s0169-5347(98)01402-5.

Cant, M.A. Social control of reproduction in banded mongooses. Anim. Behav. 2000, 59, 147–158. https://doi.org/10.1006/anbe.1999.1279.

Charpentier, M.J.; Tung, J.; Altmann, J.; Alberts, S.C. Age at maturity in wild baboons: genetic, environmental and demographic influences. Mol. Ecol. 2008, 17, 2026–2040. https://doi.org/10.1111/j.1365-294X.2008.03724.x.

Maggioncalda, A.N; Czekala, N.M.; Sapolsky, R.M. Male orangutan subadulthood: a new twist on the relationship between chronic stress and developmental arrest. Am. J. Phys. Anthropol. 2002, 118, 25–32. https://doi.org/10.1002/ajpa.10074.

Van den Berghe, F.; Paris, D.B.; Van Soom, A.; Rijsselaere, T.; Van der Weyde, L.; Bertschinger, H.J.; Paris, M.C. Reproduction in the endangered African wild dog: basic physiology, reproductive suppression and possible benefits of artificial insemination. Anim. Reprod. Sci. 2012, 133, 1–9. https://doi.org/10.1016/j.anireprosci.2012.06.003

Kruczek, M.; Styrna, J. Semen quantity and quality correlate with bank vole males' social status. Behav. Processes 2009, 82, 279–285. https://doi.org/10.1016/j.beproc.2009.07.009.

Bennett, N.C. Reproductive suppression in social Cryptomys damarensis colonies—a lifetime of socially-induced sterility in males and females (Rodentia: Bathyergidae). J. Zool. 1994, 234, 25–39. https://doi.org/10.1111/j.1469-7998.1994.tb06054.x.

Mulugeta, E.; Marion-Poll, L.; Gentien, D.; Ganswindt, S.B.; Ganswindt, A.; Bennett, N.C.; Blackburn, E.G.; Faulkes, C.G.; Heard, E. Molecular insights into the pathways underlying naked mole-rat eusociality. BioRxiv 2017, 209932. https://doi.org/10.1101/209932.

Bens, M.; Szafranski, K.; Holtze, S.; Sahm, A.; Groth, M.; Kestler, H.A.; Hildebrandt, T.B.; Platzer, M. Naked mole-rat transcriptome signatures of socially suppressed sexual maturation and links of reproduction to aging. BMC Biol. 2018, 16, 77. https://doi.org/10.1186/s12915-018-0546-z.

Sahm, A.; Platzer, M.; Koch, P.; Henning, Y.; Bens, M.; Groth, M.; Burda, H.; Begall, S.; Ting, S.; Goetz, M.; et al. Increased longevity due to sexual activity in mole-rats is associated with transcriptional changes in the HPA stress axis. Elife 2021, 10, e57843. https://doi.org/10.7554/eLife.57843.

2- The authors compared the gene expression in testes from BSB, BSA and NBS only by data from sequencing. This data should be verified by RT-PCR and WB to make sure of accuracy.

Response 2: Thank you very much for pointing out this important issue. We strongly agree with you, and it is quite necessary to verify protein expression. In the present study, we aimed to explore the molecular mechanism of gene expression in spermatogenesis of plateau zokor testes and mainly focused on the results of bioinformatics analysis. The protein expression and function analysis will be conducted in the further research.  

Anti-Müllerian hormone (AMH) is a Sertoli cell-secreted protein that plays a major role in the development of internal male genitalia. High expression of AMH in male gonads at the critical stage of embryonic genital development, i.e., 7 weeks of gestation, promotes regression of the Müllerian duct. In the absence of AMH, Müllerian ducts develop into female internal sex organs. AMH is named based on these processes (Xu et al., 2019). In the male, AMH expression begins when the seminiferous cords differentiate in the fetus (Lukas-Croisier et al., 2003), and remains high until puberty. The onset of AMH expression in fetal life is independent from gonadotropins, and involves several transcription factors. Sex-determining region Y box 9 (Sox9), steroidogenic factor-1 (SF1), and GATA factors are implicated in the transcriptional activation of AMH (Tremblay and Viger, 1999; Watanabe et al., 2000; Lasala et al., 2011). Chang et al. (2004) found that knockout of androgen receptors in Sertoli cells in mice induced a significant decrease in testosterone levels and thus gave rise to transiently elevated expression of AMH at both the mRNA and protein levels in Sertoli cells. The aforementioned studies showed that AMH protein increased with the increase of AMH mRNA expression. In the next study, we will focus on their protein expression levels.

Xu, H.Y.; Zhang, H.X.; Xiao, Z.; Qiao, J.; Li, R. Regulation of anti-Müllerian hormone (AMH) in males and the associations of serum AMH with the disorders of male fertility. Asian J. Androl. 2019, 21, 109–114. https://doi.org/10.4103/aja.aja_83_18.

Lukas-Croisier, C.; Lasala, C.; Nicaud, J.; et al. Follicle-stimulating hormone increases testicular Anti-Mullerian hormone (AMH) production through sertoli cell proliferation and a nonclassical cyclic adenosine 5'-monophosphate-mediated activation of the AMH Gene. Mol. Endocrinol. 2003, 17, 550-561. https://doi.org/10.1210/me.2002-0186.

Tremblay, J.J.; Viger, R.S. Transcription factor GATA-4 enhances Müllerian inhibiting substance gene transcription through a direct interaction with the nuclear receptor SF-1. Mol. Endocrinol. 1999, 13, 1388-1401. https://doi.org/10.1210/mend.13.8.0330.

Watanabe, K.; Clarke, T.R.; Lane, A.H.; Wang, X.; Donahoe, P.K. Endogenous expression of Müllerian inhibiting substance in early postnatal rat sertoli cells requires multiple steroidogenic factor-1 and GATA-4-binding sites. Proc. Natl. Acad. Sci. USA 2000, 97, 1624-1629. https://doi.org/10.1073/pnas.97.4.1624.

Lasala, C.; Schteingart, H.F; Arouche, N.; et al. SOX9 and SF1 are involved in cyclic AMP-mediated upregulation of anti-Mullerian gene expression in the testicular prepubertal Sertoli cell line SMAT1. Am. J. Physiol. Endocrinol. Metab. 2011, 301, E539-E547. https://doi.org/10.1152/ajpendo.00187.2011.

Chang, C.; Chen, Y.T.; Yeh, S.D.; Xu, Q.; Wang, R.S.; Guillou, F.; Lardy, H.; Yeh, S. Infertility with defective spermatogenesis and hypotestosteronemia in male mice lacking the androgen receptor in Sertoli cells. Proc. Natl. Acad. Sci. USA 2004, 101, 6876–6881. https://doi.org/10.1073/pnas.0307306101.

3- The authors analyzed the stage of spermatogenesis defects in BSA and NBS only by analyzing the sequencing data, and concluded that “spermatogenesis ceased completely in the non-breeding season testes and that the testes of the non-breeders underwent partial spermatogenesis”. However, the sequencing data is not sufficient to support the “completely” cease of spermatogenesis. They should stain some markers for spermatogonia, spermatocytes and spermatids on testis sections, and then analyze the result from the IHC staining carefully to see how spermatogenesis occurred in BSA and NBS.

Response 3: Thank you for these precious comments and suggestions. We have made the necessary changes to the relevant statement: The results suggest that spermatogenic arrest in the non-breeding season testes. We quite agree with you, and we understand that IHC staining can better reveal the gene expression of spermatogenesis in plateau zokor testes. However, in the present study, we mainly focused on the results of bioinformatics analysis and screening possible transcription factors and pathways at the transcriptome level. The result of transcriptome data may not be optimal, but should be sufficient to draw a conclusion that genes related to spermatogenesis is significantly downregulated at both meiotic and post-meiotic stages in non-breeders of plateau zokor. In the next study, we will focus on the results of  IHC staining and protein expression levels to see how spermatogenesis occurred in BSA and NBS and verify our results by RT-PCR and WB.

Minor concerns:

1- In section 3.3, the authors concluded “upregulation of apoptosis and the downregulation of spermatogenesis and meiosis suggest that spermatogenic arrest in the testes of non-breeders may occur as early as the spermatogonial stem cell differentiation stage”, However, they did not find significant difference in the mRNA levels of ID4, STRA8 and ZBTB16 between BSA, BSB and NBS. Thus, this conclusion is not well supported by their own data..

Response 1: Thank you for pointing this out. Spermatogenic arrest in the testes of non-breeders may occur at the spermatogonia cell stage.

All the males were normal individuals with different testicular sizes. Through HE staining, it can be clearly observed that the germ cells of testicular tissue of BSA plateau zokors stayed at the stage of spermatogonia, which was similar to the results of NSB plateau zokors, indicating that BSA plateau zokors were similar to NSB plateau zokors. Testes of BSA individuals degenerated, but they still had the potential to develop into mature individuals during the breeding period and participated in population reproduction.

An et al. (2020) found that in non-breeding season, seminiferous tubules only contained spermatogonia cells, indicating spermatogenesis had been arrested. STRA8 is expressed in differentiated spermatogonia and pre-meiotic spermatocytes (Ma et al., 2018). ZBTB16 is expressed in undifferentiated spermatogonia and generally used as a marker for undifferentiated spermatogonia (Fayomi and Orwig, 2018). ID-4 is a transcriptional repressor that regulates the balance between self-renewal and differentiation of spermatogonia cells (Oatley et al., 2011).

As mentioned above, STRA8, ZBTB16 and ID-4 are expressed in spermatogonia cell. Thus, there was no significant difference in mRNA levels among the BSA, BSB, and NBS for either STRA8 or ZBTB16 or ID-4. (Line 386-394).

An, X.; Wang, Y.; Li, Y.; Jia, G.; Yang, Q. Morphological features and regulation of seasonal spermatogenesis in plateau zokor (Eospalax baileyi). Acta Theriol. Sin. 2020, 40, 435–445. http://www.mammal.cn/CN/10.16829/j.slxb.150420.

Ma, H.T.; Niu, C.M.; Xia, J.; Shen, X.Y.; Xia, M.M.; Hu, Y.Q.; Zheng, Y. Stimulated by retinoic acid gene 8 (Stra8) plays important roles in many stages of spermatogenesis. Asian J. Androl. 2018, 20, 479–487. https://doi.org/10.4103/aja.aja_26_18.

Fayomi, A.P.; Orwig, K.E. Spermatogonial stem cells and spermatogenesis in mice, monkeys and men. Stem Cell Res. 2018, 29, 207–214. https://doi.org/10.1016/j.scr.2018.04.009.

Oatley, M.J.; Kaucher, A.V.; Racicot, K.E.; Oatley, J.M. Inhibitor of DNA binding 4 is expressed selectively by single spermatogonia in the male germline and regulates the self-renewal of spermatogonial stem cells in mice. Biol. Reprod. 2011, 85, 347–356. https://doi.org/10.1095/biolreprod.111.091330.

2- In fig 5, they used 3 and 2 genes as markers for pre-meiotic and post meiotic stages, but only Sycp3 was used as marker for meiotic stage. More markers should be shown for meiotic stage.

Response 2: Added. We used 2 genes as markers for each of the pre-meiotic, post meiotic and meiotic stage. (Line 186-192). 

3- Some descriptions in this paper are not appropriate. Such as: “the downregulation of spermatogenesis and meiosis”, “Meiosis was downregulated”. Meiosis and spermatogenesis can not be up or down regulated, only the expression of genes in these processes can be up or down regulated.

Response 3: Thank you for pointing this out. We have modified them. For example, in the meiosis and post-meiosis cluster, the downregulated genes in BSA-BSB and the upregulated genes in BSB-NBS were significantly enriched in GO terms related to spermatogenesis and sperm structure. (Line 203-213, 217-223).

4- Besides, there are a lot of typographical errors.

Response 4: Thank you for your pertinent recommendation. Many grammatical or typographical errors have been revised.

Reviewer 2 Report

The author Baohui Yao et al. submitted a manuscript in International Journal of Molecular Sciences (ID: ijms-2143194), entitled “Reproductive suppression caused by spermatogenic arrest: transcriptomic evidence from a non-social animal)”.

The manuscript is a study carried out on the Plateau zokor, a typical solitary rodent in the Qinghai-Tibet region. The aim is to investigate the differences in testicular size, morphology, hormonal levels, and RNA-seq transcriptome gene expression between the various samples, described in the manuscript, during spermatogenesis. In fact, spermatogenesis is a multifunctional molecular mechanism influenced by hormones, seasons and environmental pollutants, acting as endocrine disrupting chemicals. The results indicated that the principal transcript is AMH, that is also able to regulate the transcription factors SOX9, SF1, and GATA4. These transcripts significantly change during spermatogenesis in different models, at different point of development (mitotic, meiotic, post-meiotic, and somatic clusters). Another important data is the variation of testosterone levels during spermatogenesis.

All the data obtained in this study are of helpful to clarify testis development in zokors.

The paper suitable for publication in International Journal of Molecular Sciences following this minor revision:

For figure 4 and 7, please add the oligonucleotides sequences used in the RT-qPCR.

Author Response

1- For figure 4 and 7, please add the oligonucleotides sequences used in the RT-qPCR.

Response 1: There were no related oligonucleotides sequences about Figures 4 and 7. Figure 4 shows mRNA levels of AMH, SOX9, SF1, and GATA4 among BSA, BSB and NBS. Figure 7 shows KEGG enrichment of differentially expressed genes in BSA, BSB, and NBS.

Reviewer 3 Report

Dear autors

“Reproductive suppression caused by spermatogenic arrest: transcriptomic evidence from a non-social animal” is a study performed on a typical solitary subterranean rodent in the Qinghai-Tibet: the plateau zokors, considered harmful. In this species, little is known of the reproductive activity and in particular of the reproduction suppression phenomenon able of regulating the birth rate. There are more numerous papers in the literature that have studied this phenomenon in social animals, in which hierarchy plays an important role in establishing which animals can reproduce or not. Instead, the plateau zokors is a solitary animal and this work manages to give important indications concerning the functional, molecular, and biochemical aspects responsible for the reproduction suppression, both in the reproductive season and not. The increase of knowledge in this field can contribute to helping man to eventually fight effectively against the numerical increase of these animals. Many parameters were evaluated: Size of the testicles, serum hormone evaluations, analysis of the gene expression involved in spermatogenesis, mitosis, meiosis and so on, evaluation of gene clusters expressed at some stages of mouse spermatogenesis.

The study is well done, with many interesting data. Following there are some suggestions.

The Introduction is too long; I would advise to reduce the part where the authors dwell on describing the findings on the reproduction of social animals.

L. 502: How was the euthanasia carried out? Will you describe, please?ors,

“Reproductive suppression caused by spermatogenic arrest: transcriptomic evidence from a non-social animal” is a study performed on a typical solitary subterranean rodent in the Qinghai-Tibet: the plateau zokors, considered harmful. In this species, little is known of the reproductive activity and in particular of the reproduction suppression phenomenon able of regulating the birth rate. There are more numerous papers in the literature that have studied this phenomenon in social animals, in which hierarchy plays an important role in establishing which animals can reproduce or not. Instead, the plateau zokors is a solitary animal and this work manages to give important indications concerning the functional, molecular, and biochemical aspects responsible for the reproduction suppression, both in the reproductive season and not. The increase of knowledge in this field can contribute to helping man to eventually fight effectively against the numerical increase of these animals. Many parameters were evaluated: Size of the testicles, serum hormone evaluations, analysis of the gene expression involved in spermatogenesis, mitosis, meiosis and so on, evaluation of gene clusters expressed at some stages of mouse spermatogenesis.

The study is well done, with many interesting data. Following there are some suggestions.

The Introduction is too long; I would advise to reduce the part where the authors dwell on describing the findings on the reproduction of social animals.

L. 502: How was the euthanasia carried out? Will you describe it, please?

Furthermore, the animals that were captured and unused animals were used for other studies or euthanized?

Author Response

1- The Introduction is too long; I would advise to reduce the part where the authors dwell on describing the findings on the reproduction of social animals.

Response 1: Thank you for pointing this out. We have shortened the introduction and reduced unnecessary examples to streamline the narrative.

2- L. 502: How was the euthanasia carried out? Will you describe it, please?

Response 2: Plateau zokors were euthanized under anesthesia with isoflurane inhalation. (Line 471).

3- Furthermore, the animals that were captured and unused animals were used for other studies or euthanized?

Response 3: There were no unused animals in this study and all were euthanized and dissected as it was necessary to weigh the testes of all individuals. (Line 468-478).

Reviewer 4 Report

The manuscript from Yao et al. entitled "Reproductive suppression caused by spermatogenic arrest: transcriptomic evidence from a non-social animal" is a very interesting transcriptomic investigation of differentially expressed genes in testis related to the mechanism of reproductive suppression of solitary rodent plateau zokors. The manuscript is well organized and the cited literature and is relevant to the questions discussed. However, before recommending it for publication, I believe that there are still some weak points in the research, therefore the manuscript should be improved or modified.

Minor corrections:

Line 118: “NBS”, not “NBB”. Please correct it.

Line 126: Leydig cells could not be observed within the seminiferous tubule, as spermatogonia, spermatocytes, spermatids or spermatozoa because interstitial cells are located between seminiferous tubules. Please correct it.

Figure 1A, B: the letter labels on the groups are wrong according to the values (the height of the columns) and the text (lines 119-122). Please modify them.

Figure 1 and 2: for the columns, please respect the same order of the groups as in the text (BSA, BSB and NBS). Please modify them.

Table 1: Collagen type III (or 3), not iii. Please correct it.

Figure 3: the first DEG is between BSA-BSB (as in text at line 164), not BSA-BSA. Please correct it in graphic.

In figure 4 (B): the titles of the graphics must by written with capital letter (AMH, …).

Major corrections:

Serum hormone difference is very important to understand the mechanism of reproductive suppression. The authors did not give any explanation regarding the FSH, LH and testosterone level between the groups. Why the highest level of LH that was on BSA group is not reflected in the level of testosterone? Why the highest level of FSH did not stimulate spermatogenesis at the BSA group according to the histological results? The answer could be at the molecular level, and the results of transcriptomic assays are not enough discussed. I do not know why the authors discussed about puberty delay when their results were obtained on adult groups of plateau zokos? For these reasons I strongly recommend critical revision of chapter 3.

The conclusions must be focused on the title of the manuscript and on the aim of it.

Author Response

Minor corrections:

1- Line 118: “NBS”, not “NBB”. Please correct it.

Response 1: Changed. (Line 105).

2- Line 126: Leydig cells could not be observed within the seminiferous tubule, as spermatogonia, spermatocytes, spermatids or spermatozoa because interstitial cells are located between seminiferous tubules. Please correct it.

Response 2: Thank you for pointing this out. Changed. (Line 109-113).

3- Figure 1A, B: the letter labels on the groups are wrong according to the values (the height of the columns) and the text (lines 119-122). Please modify them.

Response 3: Thank you for pointing this out. Changed. (Line 114).

4- Figure 1 and 2: for the columns, please respect the same order of the groups as in the text (BSA, BSB and NBS). Please modify them.

Response 4: Changed. See Figures 1 and 2. (Line 114, 132).

5- Table 1: Collagen type III (or 3), not iii. Please correct it.

Response 5: Changed. See Table 1. (Line 152).

6- Figure 3: the first DEG is between BSA-BSB (as in text at line 164), not BSA-BSA. Please correct it in graphic.

Response 6: Thank you for pointing this out. Changed. (Line 146).

7- In figure 4 (B): the titles of the graphics must by written with capital letter (AMH, …).

Response 7: Changed. (Line 161).

Major corrections:

8- Serum hormone difference is very important to understand the mechanism of reproductive suppression. The authors did not give any explanation regarding the FSH, LH and testosterone level between the groups. Why the highest level of LH that was on BSA group is not reflected in the level of testosterone? Why the highest level of FSH did not stimulate spermatogenesis at the BSA group according to the histological results? The answer could be at the molecular level, and the results of transcriptomic assays are not enough discussed. I do not know why the authors discussed about puberty delay when their results were obtained on adult groups of plateau zokos? For these reasons I strongly recommend critical revision of chapter 3.

Response 8: Thank you for your recommendation. An et al. (2020) found that there was no significant difference in the FSH level of plateau zokor between breeding and non-breeding season, which was consistent with the results of our present study. Insufficient testosterone secretion caused spermatogenic arrest. Low level of testosterone in BSA and NBS maintain the survival of germ cells, and high level of testosterone in BSB regulate spermatogenesis and animal reproductive behavior. Although the level of LH and FSH in BSA group is high, however, the plateau zokors in BSA group were in the puberty stage, with small testicles and incomplete testicular development. In addition, genes related to meiotic cell cycle, spermatogenesis, flagellated sperm motility, fertilization, sperm capacitation, and sperm structure in BSA group were all down-regulated (Fig. 6). Therefore, the testosterone level of BSA group was low and spermatogenesis was incomplete. (Line 346-355).

All the test males were adults and normal individuals (the external genitalia and male reproductive tracts in BSB, BSA, and NBS were normal) with different testicular sizes. We have changed “delayed puberty” into “delayed testicular development” because “delayed testicular development” was more appropriate than “delayed puberty”. Delayed testicular development was manifested as small testis, low testosterone levels, and incomplete testicular development. (Line 256-291). We have made the necessary changes in the Discussion accordingly.

An, X.; Wang, Y.; Li, Y.; Jia, G.; Yang, Q. Morphological features and regulation of seasonal spermatogenesis in plateau zokor (Eospalax baileyi). Acta Theriol. Sin. 2020, 40, 435–445. http://www.mammal.cn/CN/10.16829/j.slxb.150420

9- The conclusions must be focused on the title of the manuscript and on the aim of it.

Response 9: Changed. (Line 546-552).

Round 2

Reviewer 1 Report

I have read the revised manuscript and authors' response to my concerns. I think that the authors do not understand my questions and failed to answer my concerns, particularly for Major concerns 2 and 3, as well as Minor concerns 1 and 4. Consequently, the authors did not validate their sequencing data by other means. Spermatogonial stem cells, differentiating spermatogonia and differentiated spermatogonia in the sterile animals were not identified to determine the specific stage of spermatogenesis arrested. Therefore, the revision did not allay my major concerns about the quality of the data.

Author Response

Comments and Suggestions for Authors

I have read the revised manuscript and authors' response to my concerns. I think that the authors do not understand my questions and failed to answer my concerns, particularly for Major concerns 2 and 3, as well as Minor concerns 1 and 4. Consequently, the authors did not validate their sequencing data by other means. Spermatogonial stem cells, differentiating spermatogonia and differentiated spermatogonia in the sterile animals were not identified to determine the specific stage of spermatogenesis arrested. Therefore, the revision did not allay my major concerns about the quality of the data.

Major concerns:

2- The authors compared the gene expression in testes from BSB, BSA and NBS only by data from sequencing. This data should be verified by RT-PCR and WB to make sure of accuracy.

Response 2: Thank you very much for pointing out this important issue. In order to validate the expression profile of genes from RNA-seq, we chose nine genes for further qPCR detection. The results showed that the change trends of these nine genes detected by qPCR were consistent with those from the RNA-seq data. The qPCR validation further improves the reliability of the present study (Figure 9).

3- The authors analyzed the stage of spermatogenesis defects in BSA and NBS only by analyzing the sequencing data, and concluded that “spermatogenesis ceased completely in the non-breeding season testes and that the testes of the non-breeders underwent partial spermatogenesis”. However, the sequencing data is not sufficient to support the “completely” cease of spermatogenesis. They should stain some markers for spermatogonia, spermatocytes and spermatids on testis sections, and then analyze the result from the IHC staining carefully to see how spermatogenesis occurred in BSA and NBS.

Response 3: Thank you for these precious comments and suggestions. We invesigated the morphological differences in different status plateau zokor testis using immunostaining analysis. From immunostaining analysis, we observed that PLZF (ZBTB16) and KIT were expressed in type A spermatogonia in the BSA, BSB, and NBS testis. SYCP3 was expressed in spermatocytes in the BSB testis. SYCP3 was not expressed in the BSA and NBS testis (Figure 6).

ZBTB16 is expressed in undifferentiated spermatogonia and generally used as a marker for undifferentiated spermatogonia (Fayomi and Orwig, 2018). The KIT is a marker for differentiating spermatogonial stem cells in several species including mice and goats (Jung et al., 2015). SYCP3 is required for the assembly of the conjugation complex and is expressed in human spermatocytes during prophase I of meiosis from spermatogonia to the coelomic phase (Syrjänen et al., 2014).

Fayomi, A.P.; Orwig, K.E. Spermatogonial stem cells and spermatogenesis in mice, monkeys and men. Stem Cell Res. 2018, 29, 207–214. https://doi.org/10.1016/j.scr.2018.04.009.

Jung, H.; Song, H.; Yoon, M. The KIT is a putative marker for differentiating spermatogonia in stallions. Anim. Reprod. Sci. 2015. 152, 39–46. https://doi.org/10.1016/j.anireprosci.2014.11.004.

Syrjänen, J.L.; Pellegrini, L.; Davies, O.R. A molecular model for the role of SYCP3 in meiotic chromosome organisation. Elife 2014, 3, e02963. https://doi.org/10.7554/eLife.02963.

Minor concerns:

1- In section 3.3, the authors concluded “upregulation of apoptosis and the downregulation of spermatogenesis and meiosis suggest that spermatogenic arrest in the testes of non-breeders may occur as early as the spermatogonial stem cell differentiation stage”, However, they did not find significant difference in the mRNA levels of ID4, STRA8 and ZBTB16 between BSA, BSB and NBS. Thus, this conclusion is not well supported by their own data.

Response 1: Thank you for pointing this out. We found that the mRNA levels of pre-meiotic markers (ID4, STRA8 and ZBTB16) between BSA, BSB and NBS were not significant difference. Compare with BSB, the mRNA levels of meiotic markers and post-meiotic markers were significantly decreased in BSA and NBS. Immunohistochemical results showed that ZBTB16 (PLZF) and KIT were expressed in spermatogonia in the BSA, BSB, and NBS testis. But SYCP3 only was expressed in spermatocytes in the BSB testis. Therefore, the differences in spermatogenesis in both breeder and non-breeder were mainly at the meiotic stage in the plateau zokor. Spermatogenesis in non-breeder testes is blocked during meiosis.

4- Besides, there are a lot of typographical errors.

Response 4: Thank you for your pertinent recommendation. Many grammatical or typographical errors have been revised.

Reviewer 3 Report

Dear Authors,

in the review you have responded adequately to my observations, so I approve, as far as I am concerned, the publication of the manuscript.

Author Response

Thank you for your comments.

Reviewer 4 Report

I believe that the authors have made important changes to the manuscript according to the suggestions and the manuscript looks much better at the moment and corresponds to the requirements for publication, which is why I recommend its publication in this form.

Author Response

Thank you for your comments.

Round 3

Reviewer 1 Report

The authors have tried to answer my concerns and I agree with them on what they said generally. I suggested that it can be accepted after minor revision, because the most important conlusion of this paper "Spermatogenesis in non-breeder testes is blocked during meiosis" is inferred from the expression of SYCP3 and SYCE1, not supported by Fig1C and Fig6c. The author should discuss these inconsistence carefully.